# MASS-EDITING MEMORY IN A TRANSFORMER

**Kevin Meng**[1,2]   **Arnab Sen Sharma**[2]   **Alex Andonian**[1]   **Yonatan Belinkov**[† 3]   **David Bau**[2]

[1]MIT CSAIL   [2]Northeastern University   [3]Technion – IIT

## ABSTRACT

Recent work has shown exciting promise in updating large language models with new memories, so as to replace obsolete information or add specialized knowledge. However, this line of work is predominantly limited to updating single associations. We develop MEMIT, a method for directly updating a language model with many memories, demonstrating experimentally that it can scale up to *thousands of associations* for GPT-J (6B) and GPT-NeoX (20B), exceeding prior work by orders of magnitude. Our code and data are at memit.baulab.info.

## 1   INTRODUCTION

How many memories can we add to a deep network by directly editing its weights?

Although large autoregressive language models (Radford et al., 2019; Brown et al., 2020; Wang & Komatsuzaki, 2021; Black et al., 2022) are capable of recalling an impressive array of common facts such as "Tim Cook is the CEO of Apple" or "Polaris is in the constellation Ursa Minor" (Petroni et al., 2020; Brown et al., 2020), even very large models are known to lack more specialized knowledge, and they may recall obsolete information if not updated periodically (Lazaridou et al., 2021; Agarwal & Nenkova, 2022; Liska et al., 2022). The ability to maintain fresh and customizable information is desirable in many application domains, such as question answering, knowledge search, and content generation. For example, we might want to keep search models updated with breaking news and recently-generated user feedback. In other situations, authors or companies may wish to customize models with specific knowledge about their creative work or products. Because re-training a large model can be prohibitive (Patterson et al., 2021) we seek methods that can update knowledge directly.

To that end, several *knowledge-editing* methods have been proposed to insert new memories directly into specific model parameters. The approaches include constrained fine-tuning (Zhu et al., 2020), hypernetwork knowledge editing (De Cao et al., 2021; Hase et al., 2021; Mitchell et al., 2021; 2022), and rank-one model editing (Meng et al., 2022). However, this body of work is typically limited to updating at most a few dozen facts; a recent study evaluates on a maximum of 75 (Mitchell et al., 2022) whereas others primarily focus on single-edit cases. In practical settings, we may wish to

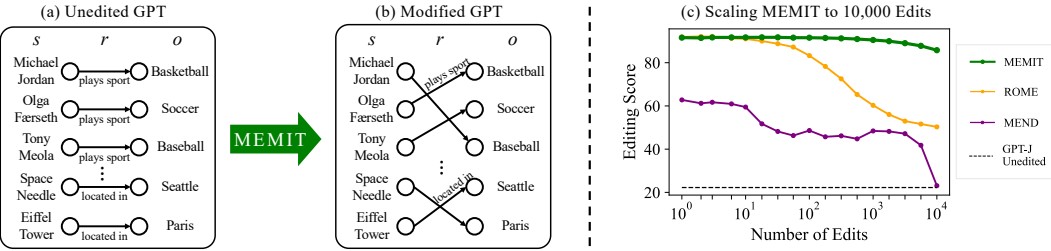

Figure 1: **MEMIT is capable of updating thousands of memories at once**. (a) Language models can be viewed as knowledge bases containing memorized tuples $(s, r, o)$, each connecting some subject $s$ to an object $o$ via a relation $r$, e.g., ($s$ = Michael Jordan, $r$ = plays sport, $o$ = basketball). (b) MEMIT modifies transformer weights to edit memories, e.g., "Michael Jordan now plays the sport baseball," while (c) maintaining generalization, specificity, and fluency at scales beyond other methods. As Section 5.2.2 details, editing score is the harmonic mean of efficacy, generalization, and specificity metrics.

[†]Supported by the Viterbi Fellowship in the Center for Computer Engineering at the Technion.
Correspondence to mengk@mit.edu, davidbau@northeastern.edu.

update a model with hundreds or thousands of facts simultaneously, but a naive sequential application of current state-of-the-art knowledge-editing methods fails to scale up (Section 5.2).

We propose MEMIT, a scalable multi-layer update algorithm that uses explicitly calculated parameter updates to insert new memories. Inspired by the ROME direct editing method (Meng et al., 2022), MEMIT targets the weights of transformer modules that we determine to be causal mediators of factual knowledge recall. Experiments on GPT-J (6B parameters; Wang & Komatsuzaki 2021) and GPT-NeoX (20B; Black et al. 2022) demonstrate that **MEMIT can scale and successfully store thousands of memories in bulk**. We analyze model behavior when inserting true facts, counterfactuals, 27 specific relations, and different mixed sets of memories. In each setting, we measure robustness in terms of generalization, specificity, and fluency while comparing the scaling of MEMIT to rank-one, hypernetwork, and fine-tuning baselines.

## 2 RELATED WORK

**Scalable knowledge bases.** The representation of world knowledge is a core problem in artificial intelligence (Richens, 1956; Minsky, 1974), classically tackled by constructing *knowledge bases* of real-world concepts. Pioneering hand-curated efforts (Lenat, 1995; Miller, 1995) have been followed by web-powered knowledge graphs (Auer et al., 2007; Bollacker et al., 2007; Suchanek et al., 2007; Havasi et al., 2007; Carlson et al., 2010; Dong et al., 2014; Vrandečić & Krötzsch, 2014; Bosselut et al., 2019) that extract knowledge from large-scale sources. Structured knowledge bases can be precisely queried, measured, and updated (Davis et al., 1993), but they are limited by sparse coverage of uncatalogued knowledge, such as commonsense facts (Weikum, 2021).

**Language models as knowledge bases.** Since LLMs can answer natural-language queries about real-world facts, it has been proposed that they could be used directly as knowledge bases (Petroni et al., 2019; Roberts et al., 2020; Jiang et al., 2020; Shin et al., 2020). However, LLM knowledge is only implicit; responses are sensitive to specific phrasings of the prompt (Elazar et al., 2021; Petroni et al., 2020), and it remains difficult to catalog, add, or update knowledge (AlKhamissi et al., 2022). Nevertheless, LLMs are promising because they scale well and are unconstrained by a fixed schema (Safavi & Koutra, 2021). In this paper, we take on the update problem, asking how the implicit knowledge encoded within model parameters can be mass-edited.

**Hypernetwork knowledge editors.** Several meta-learning methods have been proposed to edit knowledge in a model. Sinitsin et al. (2019) proposes a training objective to produce models amenable to editing by gradient descent. De Cao et al. (2021) proposes a Knowledge Editor (KE) hypernetwork that edits a standard model by predicting updates conditioned on new factual statements. In a study of KE, Hase et al. (2021) find that it fails to scale beyond a few edits, and they scale an improved objective to 10 beliefs. MEND (Mitchell et al., 2021) also adopts meta-learning, inferring weight updates from the gradient of the inserted fact. To scale their method, Mitchell et al. (2022) proposes SERAC, a system that routes rewritten facts through a different set of parameters while keeping the original weights unmodified; they demonstrate scaling up to 75 edits. Rather than meta-learning, our method employs direct parameter updates based on an explicitly computed mapping.

**Direct model editing.** Our work most directly builds upon efforts to localize and understand the internal mechanisms within LLMs (Elhage et al., 2021; Dar et al., 2022). Based on observations from Geva et al. (2021; 2022) that transformer MLP layers serve as key–value memories, we narrow our focus to them. We then employ causal mediation analysis (Pearl, 2001; Vig et al., 2020; Meng et al., 2022), which implicates a specific range of layers in recalling factual knowledge. Previously, Dai et al. (2022) and Yao et al. (2022) have proposed editing methods that alter sparse sets of neurons, but we adopt the classical view of a linear layer as an associative memory (Anderson, 1972; Kohonen, 1972). Our method is closely related to Meng et al. (2022), which also updates GPT as an explicit associative memory. Unlike the single-edit approach taken in that work, we modify a sequence of layers and develop a way for thousands of modifications to be performed simultaneously.

## 3 PRELIMINARIES: LANGUAGE MODELING AND MEMORY EDITING

The goal of MEMIT is to modify factual associations stored in the parameters of an autoregressive LLM. Such models generate text by iteratively sampling from a conditional token distribution

$\mathbb{P}\left[x_{[t]} \mid x_{[1]}, \ldots, x_{[E]}\right]$ parameterized by a $D$-layer transformer decoder, $G$ (Vaswani et al., 2017):

$$\mathbb{P}\left[x_{[t]} \mid x_{[1]}, \ldots, x_{[E]}\right] \triangleq G([x_{[1]}, \ldots, x_{[E]}]) = \mathrm{softmax}\left(W_y h_{[E]}^D\right), \tag{1}$$

where $h_{[E]}^D$ is the transformer's hidden state representation at the final layer $D$ and ending token $E$. This state is computed using the following recursive relation:

$$h_{[t]}^l(x) = h_{[t]}^{l-1}(x) + a_{[t]}^l(x) + m_{[t]}^l(x) \tag{2}$$

$$\text{where } a^l = \mathrm{attn}^l\left(h_{[1]}^{l-1}, h_{[2]}^{l-1}, \ldots, h_{[t]}^{l-1}\right) \tag{3}$$

$$m_{[t]}^l = W_{out}^l \, \sigma\left(W_{in}^l \gamma\left(h_{[t]}^{l-1}\right)\right), \tag{4}$$

$h_{[t]}^0(x)$ is the embedding of token $x_{[t]}$, and $\gamma$ is layernorm. Note that we have written attention and MLPs in parallel as done in Black et al. (2021) and Wang & Komatsuzaki (2021).

Large language models have been observed to contain many memorized facts (Petroni et al., 2020; Brown et al., 2020; Jiang et al., 2020; Chowdhery et al., 2022). In this paper, we study facts of the form (subject $s$, relation $r$, object $o$), e.g., ($s$ = Michael Jordan, $r$ = plays sport, $o$ = basketball). A generator $G$ can recall a memory for $(s_i, r_i, *)$ if we form a natural language prompt $p_i = p(s_i, r_i)$ such as "Michael Jordan plays the sport of" and predict the next token(s) representing $o_i$. Our goal is to edit many memories at once. We formally define a list of edit requests as:

$$\mathcal{E} = \{(s_i, r_i, o_i) \mid i\} \text{ s.t. } \nexists i, j. \, (s_i = s_j) \wedge (r_i = r_j) \wedge (o_i \neq o_j). \tag{5}$$

The logical constraint ensures that there are no conflicting requests. For example, we can edit Michael Jordan to play $o_i$ = "baseball", but then we exclude associating him with professional soccer.

What does it mean to edit a memory well? At a superficial level, a memory can be considered edited after the model assigns a higher probability to the statement "Michael Jordan plays the sport of baseball" than to the original prediction (basketball); we say that such an update is *effective*. Yet it is important to also view the question in terms of *generalization*, *specificity*, and *fluency*:

- To test for *generalization*, we can rephrase the question: "What is Michael Jordan's sport? What sport does he play professionally?" If the modification of $G$ is superficial and overfitted to the specific memorized prompt, such predictions will fail to recall the edited memory, "baseball."
- Conversely, to test for *specificity*, we can ask about similar subjects for which memories should not change: "What sport does Kobe Bryant play? What does Magic Johnson play?" These tests will fail if the updated $G$ indiscriminately regurgitates "baseball" for subjects that were not edited.
- When making changes to a model, we must also monitor *fluency*. If the updated model generates disfluent text such as "baseball baseball baseball baseball," we should count that as a failure.

Achieving these goals is challenging, even for a few edits (Hase et al., 2021; Mitchell et al., 2022; Meng et al., 2022). We investigate whether they can be attained at the scale of thousands of edits.

## 4 METHOD

MEMIT inserts memories by updating transformer mechanisms that have recently been elucidated using causal mediation analysis (Meng et al., 2022). In GPT-2 XL, we found that there is a sequence of critical MLP layers $\mathcal{R}$ that mediate factual association recall at the last subject token $S$ (Figure 2). MEMIT operates by (i) calculating the vector associations we want the critical layers to remember, then (ii) storing a portion of the desired memories in each layer $l \in \mathcal{R}$.

Throughout this paper, our focus will be on states representing the last subject token $S$ of prompt $p_i$, so we shall abbreviate $h_i^l = h_{[S]}^l(p_i)$. Similarly, $m_i^l$ and $a_i^l$ denote $m_{[S]}^l(p_i)$ and $a_{[S]}^l(p_i)$.

### 4.1 IDENTIFYING THE CRITICAL PATH OF MLP LAYERS

Figure 3 shows the results of applying causal tracing to the larger GPT-J (6B) model; for implementation details, see Appendix A. We measure the average indirect causal effect of each $h_i^l$ on a sample of memory prompts $p_i$, with either the Attention or MLP modules for token $S$ disabled. The results

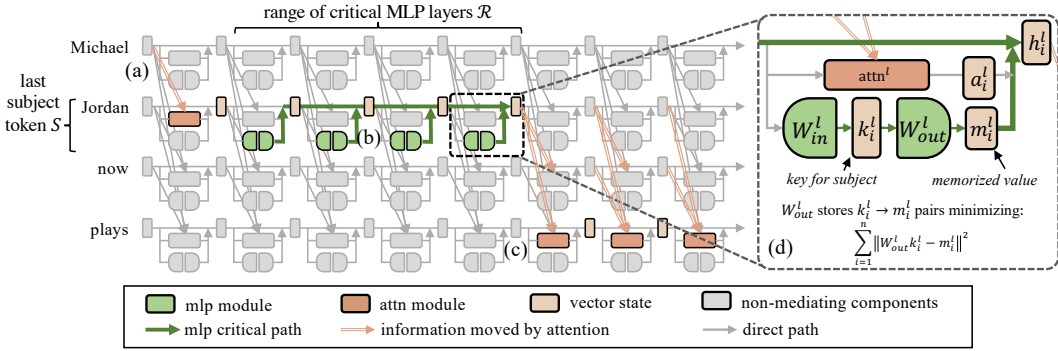

Figure 2: **MEMIT modifies transformer parameters on the critical path of MLP-mediated factual recall.** We edit stored associations based on observed patterns of causal mediation: (a) first, the early-layer attention modules gather subject names into vector representations at the last subject token $S$. (b) Then MLPs at layers $l \in \mathcal{R}$ read these encodings and add memories to the residual stream. (c) Those hidden states are read by attention to produce the output. (d) MEMIT edits memories by storing vector associations in the critical MLPs.

confirm that GPT-J has a concentration of mediating states $h_i^l$; moreover, they highlight a mediating causal role for a range of MLP modules, which can be seen as a large gap between the effect of single states (purple bars in Figure 3) and the effects with MLP severed (green bars); this gap diminishes after layer 8. Unlike Meng et al. (2022) who use this test to identify a single edit layer, we select the whole range of critical MLP layers $l \in \mathcal{R}$. For GPT-J, we have $\mathcal{R} = \{3, 4, 5, 6, 7, 8\}$.

Given that a *range* of MLPs play a joint mediating role in recalling facts, we ask: what is the role of *one* MLP in storing a memory? Each token state in a transformer is part of the residual stream that all attention and MLP modules read from and write to (Elhage et al., 2021). Unrolling Eqn. 2 for $h_i^L = h_{[S]}^L(p_i)$:

$$h_i^L = h_i^0 + \sum_{l=1}^{L} a_i^l + \sum_{l=1}^{L} m_i^l. \quad (6)$$

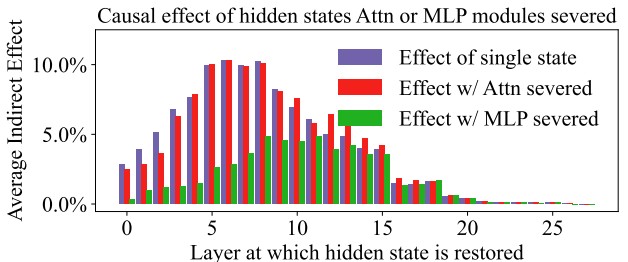

Figure 3: A critical mediating role for mid-layer MLPs.

Eqn. 6 highlights that each individual MLP contributes by *adding* to the memory at $h_i^L$ (Figure 2b), which is later read by last-token attention modules (Figure 2c). Therefore, when writing new memories into $G$, we can spread the desired changes across all the critical layers $m_i^l$ for $l \in \mathcal{R}$.

## 4.2 BATCH UPDATE FOR A SINGLE LINEAR ASSOCIATIVE MEMORY

In each individual layer $l$, we wish to store a large batch of $u \gg 1$ memories. This section derives an optimal single-layer update that minimizes the squared error of memorized associations, assuming that the layer contains previously-stored memories that should be preserved. We denote $W_0 \triangleq W_{out}^l$ (Eqn. 4, Figure 2) and analyze it as a linear associative memory (Kohonen, 1972; Anderson, 1972) that associates a set of input keys $k_i \triangleq k_i^l$ (encoding subjects) to corresponding memory values $m_i \triangleq m_i^l$ (encoding memorized properties) with minimal squared error:

$$W_0 \triangleq \underset{\hat{W}}{\operatorname{argmin}} \sum_{i=1}^{n} \left\| \hat{W} k_i - m_i \right\|^2. \quad (7)$$

If we stack keys and memories as matrices $K_0 = [k_1 \mid k_2 \mid \cdots \mid k_n]$ and $M_0 = [m_1 \mid m_2 \mid \cdots \mid m_n]$, then Eqn. 7 can be optimized by solving the normal equation (Strang, 1993, Chapter 4):

$$W_0 K_0 K_0^T = M_0 K_0^T. \quad (8)$$

Suppose that pre-training sets a transformer MLP's weights to the optimal solution $W_0$ as defined in Eqn. 8. Our goal is to update $W_0$ with some small change $\Delta$ that produces a new matrix $W_1$ with

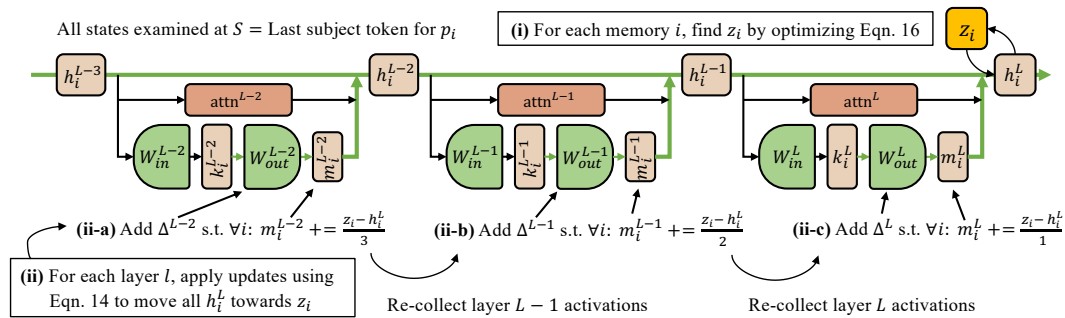

Figure 4: **The MEMIT update**. We first (i) replace $h_i^l$ with the vector $z_i$ and optimize Eqn. 16 so that it conveys the new memory. Then, after all $z_i$ are calculated we (ii) iteratively insert a fraction of the residuals for all $z_i$ over the range of critical MLP modules, executing each layer's update by applying Eqn. 14. Because changing one layer will affect activations of downstream modules, we recollect activations after each iteration.

a set of additional associations. Unlike Meng et al. (2022), we cannot solve our problem with a constraint that adds only a single new association, so we define an expanded objective:

$$W_1 \triangleq \underset{\hat{W}}{\operatorname{argmin}} \left( \sum_{i=1}^{n} \left\| \hat{W} k_i - m_i \right\|^2 + \sum_{i=n+1}^{n+u} \left\| \hat{W} k_i - m_i \right\|^2 \right). \tag{9}$$

We can solve Eqn. 9 by again applying the normal equation, now written in block form:

$$W_1 \begin{bmatrix} K_0 & K_1 \end{bmatrix} \begin{bmatrix} K_0 & K_1 \end{bmatrix}^T = \begin{bmatrix} M_0 & M_1 \end{bmatrix} \begin{bmatrix} K_0 & K_1 \end{bmatrix}^T \tag{10}$$

which expands to: $\quad (W_0 + \Delta)(K_0 K_0^T + K_1 K_1^T) = M_0 K_0^T + M_1 K_1^T \tag{11}$

$$W_0 K_0 K_0^T + W_0 K_1 K_1^T + \Delta K_0 K_0^T + \Delta K_1 K_1^T = M_0 K_0^T + M_1 K_1^T \tag{12}$$

subtracting Eqn. 8 from Eqn. 12: $\quad \Delta(K_0 K_0^T + K_1 K_1^T) = M_1 K_1^T - W_0 K_1 K_1^T. \tag{13}$

A succinct solution can be written by defining two additional quantities: $C_0 \triangleq K_0 K_0^T$, a constant proportional to the uncentered covariance of the pre-existing keys, and $R \triangleq M_1 - W_0 K_1$, the residual error of the new associations when evaluated on old weights $W_0$. Then Eqn. 13 can be simplified as:

$$\Delta = R K_1^T (C_0 + K_1 K_1^T)^{-1}. \tag{14}$$

Since pretraining is opaque, we do not have access to $K_0$ or $M_0$. Fortunately, computing Eqn. 14 only requires an aggregate statistic $C_0$ over the previously stored keys. We assume that the set of previously memorized keys can be modeled as a random sample of inputs, so that we can compute

$$C_0 = \lambda \cdot \mathbb{E}_k \left[ k k^T \right] \tag{15}$$

by estimating $\mathbb{E}_k \left[ k k^T \right]$, an uncentered covariance statistic collected using an empirical sample of vector inputs to the layer. We must also select $\lambda$, a hyperparameter that balances the weighting of new v.s. old associations; a typical value is $\lambda = 1.5 \times 10^4$.

### 4.3 Updating multiple layers

We now define the overall update algorithm (Figure 4). Inspired by the observation that robustness is improved when parameter change magnitudes are minimized (Zhu et al., 2020), we spread updates evenly over the range of mediating layers $\mathcal{R}$. We define a target layer $L \triangleq \max(\mathcal{R})$ at the end of the mediating layers, at which the new memories should be fully represented. Then, for each edit $(s_i, r_i, o_i) \in \mathcal{E}$, we (i) compute a hidden vector $z_i$ to replace $h_i^L$ such that adding $\delta_i \triangleq z_i - h_i^L$ to the hidden state at layer $L$ and token $T$ will completely convey the new memory. Finally, one layer at a time, we (ii) modify the MLP at layer $l$, so that it contributes an approximately-equal portion of the change $\delta_i$ for each memory $i$.

**(i) Computing $z_i$.** For the $i$th memory, we first compute a vector $z_i$ that would encode the association $(s_i, r_i, o_i)$ if it were to replace $h_i^L$ at layer $L$ at token $S$. We find $z_i = h_i^L + \delta_i$ by optimizing the residual vector $\delta_i$ using gradient descent:

$$z_i = h_i^L + \underset{\delta_i}{\operatorname{argmin}} \frac{1}{P} \sum_{j=1}^{P} - \log \mathbb{P}_{G(h_i^L += \delta_i)} \left[ o_i \mid x_j \oplus p(s_i, r_i) \right]. \tag{16}$$

In words, we optimize $\delta_i$ to maximize the model's prediction of the desired object $o_i$, given a set of factual prompts $\{x_j \oplus p(s_i, r_i)\}$ that concatenate random prefixes $x_j$ to a templated prompt to aid generalization across contexts. $G(h_i^L += \delta_i)$ indicates that we modify the transformer execution by substituting the modified hidden state $z_i$ for $h_i^L$; this is called "hooking" in popular ML libraries.

**(ii) Spreading $z_i - h_i^L$ over layers**. We seek delta matrices $\Delta^l$ such that:

$$\text{setting } \hat{W}_{out}^l := W_{out}^l + \Delta^l \text{ for all } l \in \mathcal{R} \text{ optimizes } \min_{\{\Delta^l\}} \sum_i \left\| z_i - \hat{h}_i^L \right\|^2, \tag{17}$$

$$\text{where } \hat{h}_i^L = h_i^0 + \sum_{l=1}^{L} a_i^l + \sum_{l=1}^{L} \hat{W}_{out}^l \, \sigma \left( W_{in}^l \gamma \left( h_t^{l-1} \right) \right). \tag{18}$$

Because edits to any layer will influence all following layers' activations, we calculate $\Delta^l$ iteratively in ascending layer order (Figure 4ii-a,b,c). To compute each individual $\Delta^l$, we need the corresponding keys $K^l = [k_1^l \mid \cdots \mid k_n^l]$ and memories $M^l = [m_1^l \mid \cdots \mid m_n^l]$ to insert using Eqn. 14. Each key $k_i^l$ is computed as the input to $W_{out}^l$ at each layer $l$ (Figure 2d):

$$k_i^l = \frac{1}{P} \sum_{j=1}^{P} k(x_j + s_i), \text{ where } k(x) = \sigma \left( W_{in}^l \, \gamma \left( h_i^{l-1}(x) \right) \right). \tag{19}$$

$m_i^l$ is then computed as the sum of its current value and a fraction of the remaining top-level residual:

$$m_i^l = W_{out} k_i^l + r_i^l \text{ where } r_i^l \text{ is the residual given by } \frac{z_i - h_i^L}{L - l + 1}, \tag{20}$$

where the denominator of $r_i$ spreads the residual out evenly. Algorithm 1 summarizes MEMIT, and additional implementation details are offered in Appendix B.

---

**Algorithm 1:** The MEMIT Algorithm

**Data:** Requested edits $\mathcal{E} = \{(s_i, r_i, o_i)\}$, generator $G$, layers to edit $\mathcal{S}$, covariances $C^l$
**Result:** Modified generator containing edits from $\mathcal{E}$

1 **for** $s_i, r_i, o_i \in \mathcal{E}$ **do**  // Compute target $z_i$ vectors for every memory $i$
2     **optimize** $\delta_i \leftarrow \text{argmin}_{\delta_i} \frac{1}{P} \sum_{j=1}^{P} - \log \mathbb{P}_{G(h_i^L += \delta_i)} [o_i \mid x_j \oplus p(s_i, r_i)]$ (Eqn. 16)
3     $z_i \leftarrow h_i^L + \delta_i$
4 **end**
5 **for** $l \in \mathcal{R}$ **do**  // Perform update: spread changes over layers
6     $h_i^l \leftarrow h_i^{l-1} + a_i^l + m_i^l$ (Eqn. 2)  // Run layer $l$ with updated weights
7     **for** $s_i, r_i, o_i \in \mathcal{E}$ **do**
8        $k_i^l \leftarrow k_i^l = \frac{1}{P} \sum_{j=1}^{P} k(x_j + s_i)$ (Eqn. 19)
9        $r_i^l \leftarrow \frac{z_i - h_i^L}{L - l + 1}$ (Eqn. 20)  // Distribute residual over remaining layers
10     **end**
11     $K^l \leftarrow [k_i^{l_1}, ..., k_i^L]$
12     $R^l \leftarrow [r_i^{l_1}, ..., r_i^L]$
13     $\Delta^l \leftarrow R^l K^{l^T} (C^l + K^l K^{l^T})^{-1}$ (Eqn. 14)
14     $W^l \leftarrow W^l + \Delta^l$  // Update layer $l$ MLP weights in model
15 **end**

---

## 5 Experiments

### 5.1 Models and baselines

We run experiments on two autoregressive LLMs: GPT-J (6B) and GPT-NeoX (20B). For baselines, we first compare with a naive fine-tuning approach that uses weight decay to prevent forgetfulness (**FT-W**). Next, we experiment with **MEND**, a hypernetwork-based model editing approach that edits multiple facts at the same time (Mitchell et al., 2021). Finally, we run a sequential version of **ROME** (Meng et al., 2022): a direct model editing method that iteratively updates one fact at a time. The recent SERAC model editor (Mitchell et al., 2022) does not yet have public code, so we cannot compare with it at this time. See Appendix B for implementation details.

## 5.2 MEMIT SCALING

### 5.2.1 EDITING 10K MEMORIES IN ZSRE

We first test MEMIT on zsRE (Levy et al., 2017), a question-answering task from which we extract 10,000 real-world facts; zsRE tests MEMIT's ability to add *correct* information. Because zsRE does not contain generation tasks, we evaluate solely on prediction-based metrics. **Efficacy**

Table 1: 10,000 zsRE Edits on GPT-J (6B).

| Editor | Score ↑ | Efficacy ↑ | Paraphrase ↑ | Specificity ↑ |
|--------|---------|-----------|--------------|---------------|
| GPT-J | 26.4 | 26.4 (±0.6) | 25.8 (±0.5) | 27.0 (±0.5) |
| FT-W | 42.1 | 69.6 (±0.6) | 64.8 (±0.6) | 24.1 (±0.5) |
| MEND | 20.0 | **19.4 (±0.5)** | **18.6 (±0.5)** | 22.4 (±0.5) |
| ROME | **2.6** | **21.0 (±0.7)** | **19.6 (±0.7)** | **0.9 (±0.1)** |
| MEMIT | **50.7** | **96.7 (±0.3)** | **89.7 (±0.5)** | **26.6 (±0.5)** |

measures the proportion of cases where $o$ is the argmax generation given $p(s, r)$, **Paraphrase** is the same metric but applied on paraphrases, **Specificity** is the model's argmax accuracy on a randomly-sampled unrelated fact that should not have changed, and **Score** is the harmonic mean of the three aforementioned scores; Appendix C contains formal definitions. As Table 1 shows, MEMIT performs best at 10,000 edits; most memories are recalled with generalization and minimal bleedover. Interestingly, simple fine-tuning FT-W performs better than the baseline knowledge editing methods MEND and ROME at this scale, likely because its objective is applied only once.

### 5.2.2 COUNTERFACT SCALING CURVES

Next, we test MEMIT's ability to add *counterfactual* information using COUNTERFACT, a collection of 21,919 factual statements (Meng et al. (2022), Appendix C). We first filter conflicts by removing facts that violate the logical condition in Eqn. 5 (i.e., multiple edits modify the same $(s, r)$ prefix to different objects). For each problem size $n \in \{1, 2, 3, 6, 10, 18, 32, 56, 100, 178, 316, 562, 1000, 1778, 3162, 5623, 10000\}$[1], $n$ counterfactuals are inserted.

Following Meng et al. (2022), we report several metrics designed to test editing desiderata. **Efficacy Success** (**ES**) evaluates editing success and is the proportion of cases for which the new object $o_i$'s probability is greater than the probability of the true real-world object $o_i^c$:[2] $\mathbb{E}_i \left[ \mathbb{P}_G \left[ o_i \mid p(s_i, r_i) \right] > \mathbb{P}_G \left[ o_i^c \mid p(s_i, r_i) \right] \right]$. **Paraphrase Success** (**PS**) is a generalization measure defined similarly, except $G$ is prompted with rephrasings of the original statement. For testing specificity, **Neighborhood Success** (**NS**) is defined similarly, but we check the probability $G$ assigns to the correct answer $o_i^c$ (instead of $o_i$), given prompts about distinct but semantically-related subjects (instead of $s_i$). **Editing Score** (**S**) aggregates metrics by taking the harmonic mean of ES, PS, NS.

We are also interested in measuring generation quality of the updated model. First, we check that $G$'s generations are semantically consistent with the new object using a **Reference Score** (**RS**), which is collected by generating text about $s$ and checking its TF-IDF similarity with a reference Wikipedia text about $o$. To test for fluency degradation due to excessive repetition, we measure **Generation Entropy** (**GE**), computed as the weighted sum of the entropy of bi- and tri-gram $n$-gram distributions of the generated text. See Appendix C for further details on metrics.

Figure 5 plots performance v.s. number of edits on log scale, up to 10,000 facts. ROME performs well up to $n = 10$ but degrades starting at $n = 32$. Similarly, MEND performs well at $n = 1$ but rapidly declines at $n = 6$, losing all efficacy before $n = 1,000$ and, curiously, having negligible effect on the model at $n = 10,000$ (the high specificity score is achieved by leaving the model nearly unchanged). MEMIT performs best at large $n$. At small $n$, ROME achieves better generalization at the cost of slightly lower specificity, which means that ROME's edits are more robust under rephrasings, likely due to that method's hard equality constraint for weight updates, compared to MEMIT's soft error minimization. Table 2 provides a direct numerical comparison at 10,000 edits on both GPT-J and GPT-NeoX. FT-W[3] does well on probability-based metrics but suffers from complete generation failure, indicating significant model damage.

Appendix B provides a runtime analysis of all four methods on 10,000 edits. We find that MEND is fastest, taking 98 sec. FT is second at around 29 min, while MEMIT and ROME are the slowest at

---

[1] These values come from a log-scale curve: $n_i = \exp\left(\ln(10,000) * \frac{i}{16}\right)$, for non-negative integers $i$.

[2] COUNTERFACT is derived from a set of true facts from WikiData, so $o_i^c$ is always known.

[3] We find that the weight decay hyperparameter is highly sensitive to the number of edits. Therefore, to evaluate scaling behavior cost-efficiently, we tune it only on $n = 10,000$. See Appendix B.1 for experimental details.

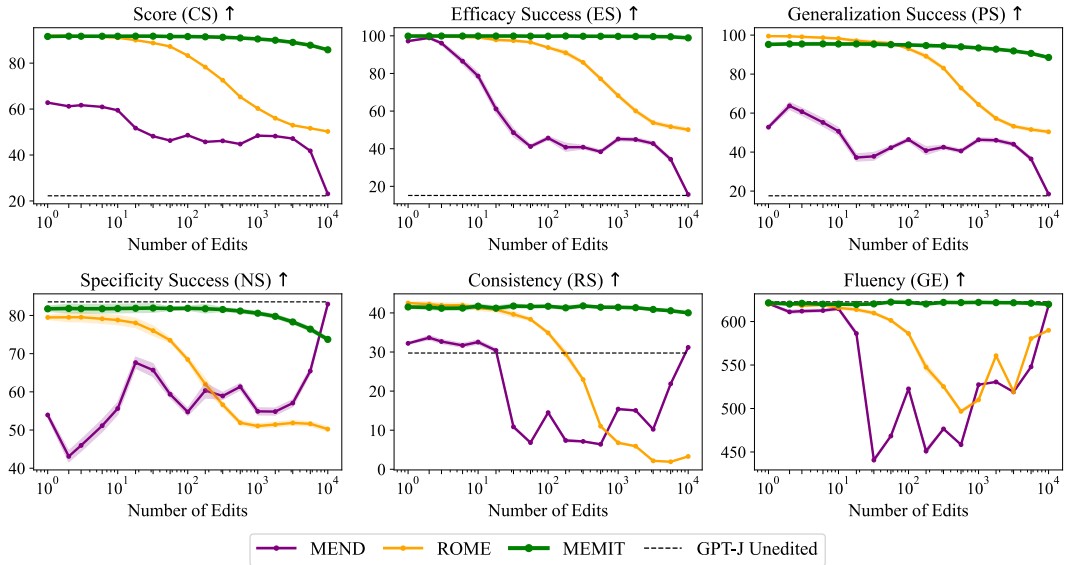

Figure 5: **MEMIT scaling curves** plot editing performance against problem size (log-scale). The dotted line indicates GPT-J's pre-edit performance; specificity (NS) and fluency (GE) should stay close to the baseline. 95% confidence intervals are shown as areas.

Table 2: Numerical results on COUNTERFACT for 10,000 edits.

| Editor | Score | Efficacy | Generalization | Specificity | Fluency | Consistency |
|---|---|---|---|---|---|---|
| | S ↑ | ES ↑ | PS ↑ | NS ↑ | GE ↑ | RS ↑ |
| GPT-J | 22.4 | 15.2 (0.7) | 17.7 (0.6) | 83.5 (0.5) | 622.4 (0.3) | 29.4 (0.2) |
| FT-W | 67.6 | **99.4 (0.1)** | 77.0 (0.7) | **46.9 (0.6)** | **293.9 (2.4)** | **15.9 (0.3)** |
| MEND | **23.1** | **15.7 (0.7)** | **18.5 (0.7)** | **83.0 (0.5)** | 618.4 (0.3) | 31.1 (0.2) |
| ROME | 50.3 | **50.2 (1.0)** | **50.4 (0.8)** | 50.2 (0.6) | **589.6 (0.5)** | **3.3 (0.0)** |
| MEMIT | **85.8** | 98.9 (0.2) | **88.6 (0.5)** | 73.7 (0.5) | **619.9 (0.3)** | **40.1 (0.2)** |
| GPT-NeoX | 23.7 | 16.8 (1.9) | 18.3 (1.7) | 81.6 (1.3) | 620.4 (0.6) | 29.3 (0.5) |
| MEMIT | 82.0 | 97.2 (0.8) | 82.2 (1.6) | 70.8 (1.4) | 606.4 (1.0) | 36.9 (0.6) |

$7.44\,\mathrm{hr}$ and $12.29\,\mathrm{hr}$, respectively. While MEMIT's execution time is high relative to MEND and FT, we note that its current implementation is naive and does not batch the independent $z_i$ optimizations, instead computing each one in series. These computations are actually "embarrassingly parallel" and thus could be batched.

## 5.3 EDITING DIFFERENT CATEGORIES OF FACTS

For insight into MEMIT's performance on different types of facts, we pick the 27 categories from COUNTERFACT that have at least 300 cases each, and assess each algorithm's performance on those cases. Figure 6a shows that MEMIT achieves better overall scores compared to FT and MEND in all categories. It also reveals that some relations are harder to edit compared to others; for example, each of the editing algorithms faced difficulties in changing the sport an athlete plays. Even on harder cases, MEMIT outperforms other methods by a clear margin.

Model editing methods are known to occasionally suffer from a trade-off between attaining high generalization and good specificity. This trade-off is clearly visible for MEND in Figure 6b. FT consistently fails to achieve good specificity. Overall, MEMIT achieves a higher score in both dimensions, although it also exhibits a trade-off in editing some relations such as P127 ("product owned by company") and P641 ("athlete plays sport").

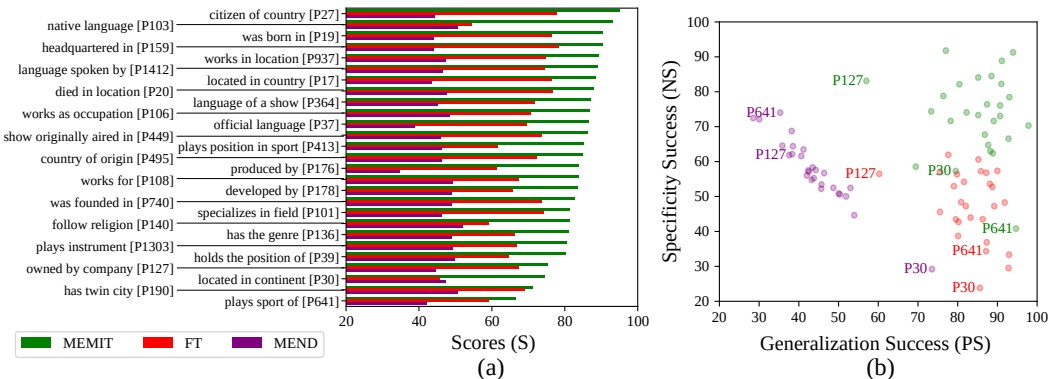

Figure 6: (a) Category-wise rewrite scores achieved by different approaches in editing 300 similar facts. (b) Category-wise *specificity* vs *generalization* scores by different approaches on 300 edits.

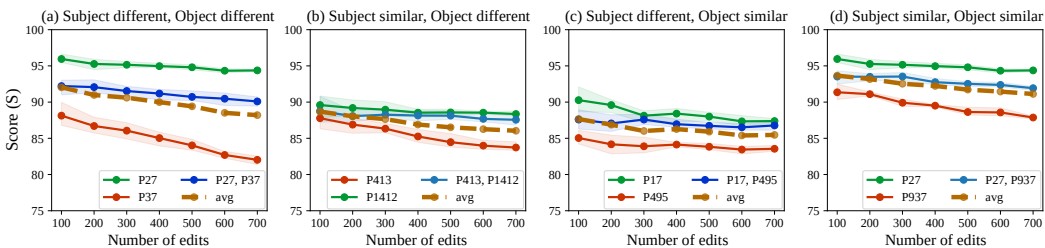

Figure 7: When comparing mixes of edits, MEMIT gives consistent near-linear (near-average) performance while scaling up to 700 facts.

## 5.4 EDITING DIFFERENT CATEGORIES OF FACTS TOGETHER

To investigate whether the scaling of MEMIT is sensitive to differences in the diversity of the memories being edited together, we sample sets of cases $\mathcal{E}_{mix}$ that mix two different relations from the COUNTERFACT dataset. We consider four scenarios depicted in Figure 7, where the relations have similar or different classes of subjects or objects. In all of the four cases, MEMIT's performance on $\mathcal{E}_{mix}$ is close to the average of the performance of each relation without mixing. This provides support to the hypothesis that the scaling of MEMIT is neither positively nor negatively affected by the diversity of the memories being edited. Appendix D contains implementation details.

## 6 DISCUSSION AND CONCLUSION

We have developed MEMIT, a method for editing factual memories in large language models by directly manipulating specific layer parameters. Our method scales to much larger sets of edits (100x) than other approaches while maintaining excellent specificity, generalization, and fluency.

Our investigation also reveals some challenges: certain relations are more difficult to edit with robust specificity, yet even on challenging cases we find that MEMIT outperforms other methods by a clear margin. The knowledge representation we study is also limited in scope to working with directional $(s, r, o)$ relations: it does not cover spatial or temporal reasoning, mathematical knowledge, linguistic knowledge, procedural knowledge, or even symmetric relations. For example, the association that "Tim Cook is CEO of Apple" must be processed separately from the opposite association that "The CEO of Apple is Tim Cook."

Despite these limitations, it is noteworthy that large-scale model updates can be constructed using an explicit analysis of internal computations. Our results raise a question: might interpretability-based methods become a commonplace alternative to traditional opaque fine-tuning approaches? Our positive experience brings us optimism that further improvements to our understanding of network internals will lead to more transparent and practical ways to edit, control, and audit models.

## 7 ETHICAL CONSIDERATIONS

Although we test a language model's ability to serve as a knowledge base, we do not find these models to be a reliable source of knowledge, and we caution readers that a LLM should not be used as an authoritative source of facts. Our memory-editing methods shed light on the internal mechanisms of models and potentially reduce the cost and energy needed to fix errors in a model, but the same methods might also enable a malicious actor to insert false or damaging information into a model that was not originally present in the training data.

## 8 ACKNOWLEDGEMENTS.

Thanks to Jaden Fiotto-Kaufmann for building the demonstration at memit.baulab.us. This project was supported by an AI Alignment grant from Open Philanthropy. YB was also supported by the Israel Science Foundation (grant No. 448/20) and an Azrieli Foundation Early Career Faculty Fellowship.

## 9 REPRODUCIBILITY

The code and data for our methods and experiments are available at memit.baulab.info.

All experiments are run on workstations with NVIDIA A6000 GPUs. The language models are loaded using HuggingFace Transformers (Wolf et al., 2019), and PyTorch (Paszke et al., 2019) is used for executing the model editing algorithms on GPUs.

GPT-J experiments fit into one 48GB A6000, but GPT-NeoX runs require at least two: one 48GB GPU for running the model in `float16`, and another slightly smaller GPU for executing the editing method. Due to the size of these language models, our experiments will not run on GPUs with less memory.

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

# A    CAUSAL TRACING

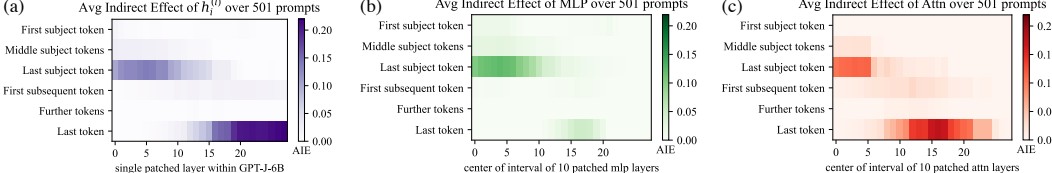

Figure 8: **Causal Tracing** (using the method of Meng et al. 2022). Each grid cell's intensity reflects the average causal indirect effect of a hidden state on the expression of a factual association, with strong causal mediators highlighted with darker colors. We find that MLPs at the last subject token and attention modules at the last token are important. The presence of influential attention activations at the earliest layers of the last subject token is investigated with additional path dependent experiments (Figure 3).

MEMIT begins by identifying MLP layers that are causal mediators for recall of factual associations in the model. To do so in GPT-J, we use code provided by Meng et al. (2022): beginning with a sample of 501 true statements of facts that are correctly predicted by GPT-J, we measure baseline predicted probabilities of each true fact when noise is introduced into encoding of the subject tokens to degrade the accuracy of the model. Then in Figure 8 (a) for each individual $h_t^l$, we restore the state to the value that it would have had without injected noise, and we plot the average improvement of predicted probability. As in Meng et al. (2022), we use Gaussian noise with standard deviation $3\sigma$ ($\sigma^2$ is the empirically observed variance of embedding activations) and plot averages for all 501 statements over 10 noise samples. For (b) and (c) we use the same procedure, except we restore runs of 10 layers of MLP outputs $m_t^l$ and 10 layers of Attn $a_t^l$, instead of full hidden states.

These measurements confirm that GPT-J has a causal structure that is similar to the structure reported by Meng et al. (2022) in their study of GPT2-XL. Unlike with GPT-XL, a strong causal effect is observed in the earliest layers of Attention at the last subject token, which likely reflects a concentrated attention computation when GPT-J is recognizing and chunking the n-gram subject name, but the path-dependent experiment (Figure 3) suggests that Attention is not an important mediator of factual recall of memories about the subject.

In the main paper, Figure 3 plots the same data as Figure 8 (a) as a bar graph, focused on only the last subject token, and it adds two additional measurements. In red bars, it repeats the measurement of causal effects of states with Attention modules at the last subject token frozen in the corrupted state, so that cannot be influenced by the state being probed, and in green bars it repeats the experiment with the MLP modules at the last subject token similarly frozen, so they cannot be influenced by the causal probe. Severing the Attention modules does not shift the curve, which suggests that Attention computations do not play a decisive mediating role in knowledge recall at the last subject token. In contrast, severing the MLP modules reveals a large gap, which suggests that, at layers where the gap is largest, the role of the MLP computation is important. We select the layers where the gap is largest as the range $\mathcal{R}$ to use for the intervention done by MEMIT.

# B    IMPLEMENTATION DETAILS

## B.1    FINE-TUNING WITH WEIGHT DECAY

Our fine-tuning baseline updates layer 21 of GPT-J, which Meng et al. (2022) found to provide the best performance in the single-edit case. Rather than using a hard $L_\infty$-norm constraint, we use a soft weight decay regularizer. However, the optimal amount of regularization depends strongly on the number of edits (more edits require higher-norm edits), so we tune this hyperparameter for the $n = 10,000$ case. Figure 9 shows that $5 \times 10^{-4}$ selects for the optimal tradeoff between generalization and specificity. FT-W optimization proceeds for a maximum of 25 steps with a learning rate of $5 \times 10^{-4}$. To prevent overfitting, early stopping is performed when the loss reaches $10^{-2}$. Regarding runtime, FT takes $1,716.21\,\text{sec} \approx 0.48\,\text{hr}$ to execute 10,000 edits on GPT-J.

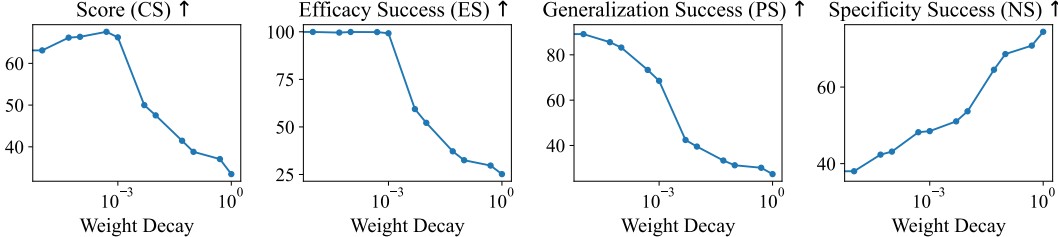

Figure 9: **Optimizing fine-tuning weight decay on 10,000 edits**. We find an evident tradeoff between generalization and specificity, opting for the value with the highest Score.

Note that we choose not to complicate the analysis by tuning FT-W on more than one layer. Table 2 demonstrates that FT-W, with just one layer, already gets near-perfect efficacy at the cost of low specificity, which indicates sufficient edit capacity.

### B.2    MODEL EDITING NETWORKS WITH GRADIENT DECOMPOSITION (MEND)

MEND makes concurrent edits by accumulating gradients from all edit examples, then passing them through the hypernetwork together. We use the GPT-J MEND hypernetwork trained by Meng et al. (2022). During inference, learning rate scale is set to the default value of 1.0. MEND is by far the fastest method, taking $98.25$ seconds to execute 10,000 updates on GPT-J.

### B.3    RANK-ONE MODEL EDITING (ROME)

The default ROME hyperparameters are available in their open source code: GPT-J updates are executed at layer 5, where optimization proceeds for 20 steps with a weight decay of 0.5, KL factor of 0.0625, and learning rate of $5 \times 10^{-1}$. ROME uses prefix sampling, resulting in 10 prefixes of length 5 and 10 prefixes of length 10. Covariance statistics are collected in `fp32` on Wikitext using a sample size of 100,000. See Meng et al. (2022) for more details.

ROME takes $44,248.26 \, \text{sec} \approx 12.29 \, \text{hr}$ for 10,000 edits on GPT-J, which works out to approximately 4 seconds per edit.

### B.4    MASS-EDITING MEMORY IN A TRANSFORMER (MEMIT)

On GPT-J, we choose $\mathcal{R} = \{3, 4, 5, 6, 7, 8\}$ and set $\lambda$, the covariance adjustment factor, to $15,000$. Similar to ROME, covariance statistics are collected using 100,000 samples of Wikitext in `fp32`. $\delta_i$ optimization proceeds for 25 steps with a learning rate of $5 \times 10^{-1}$. In practice, we clamp the $L_2$ norm of $\delta_i$ such that it is less than $\frac{3}{4}$ of the original hidden state norm, $\|h_i^L\|$. On GPT-NeoX, we select $\mathcal{R} = \{6, 7, 8, 9, 10\}$ and set $\lambda = 20,000$. Covariance statistics are collected over 50,000 samples of Wikitext in `fp16` but stored in `fp32`. Optimization for $\delta_i$ proceeds for 20 steps using a learning rate of $5 \times 10^{-1}$ while clamping $\|h_i^L\|$ to $\frac{3}{10}\|h_i^L\|$.

In MEMIT, we have the luxury of being able to pre-compute and cache $z_i$ values, since they are inserted in parallel. If all such vectors are already computed, MEMIT takes $3,226.35 \, \text{sec} \approx 0.90 \, \text{hr}$ for 10,000 updates on GPT-J, where the most computationally expensive step is inverting a large square matrix (Eqn. 14). Computing each $z_i$ vector is slightly less expensive than computing a ROME update; to get all 10,000 $z_i$ vectors, we need $23,546.65 \, \text{sec} \approx 6.54 \, \text{hr}$. This optimization is currently done in series, but it is actually "embarrassingly parallel," as we can greatly reduce computation time by batching the gradient descent steps. Note that this speed-up does not apply to ROME, since each update must be done iteratively.

## C  EVALUATION METRICS

### C.1  FOR ZSRE

For consistency with previous works that use the zsRE task (Mitchell et al., 2021; Meng et al., 2022), we report the same three probability tests:

- **Efficacy** is the proportion of edits that $G$ recalls with top-1 accuracy. Note that the prompt matches exactly what the edit method sees at runtime:

$$\mathbb{E}_i \left[ o_i = \operatorname*{argmax}_{x_E} \mathbb{P}_G \left[ x_E \mid p(s_i, r_i) \right] \right]. \tag{21}$$

- **Paraphrase** is the accuracy on rephrasings of the original statement:

$$\mathbb{E}_i \left[ \mathbb{E}_{p \in \text{paraphrases}(s_i, r_i)} \left[ o_i = \operatorname*{argmax}_{x_E} \mathbb{P}_G \left[ x_E \mid p \right] \right] \right]. \tag{22}$$

- **Specificity** is the proportion of neighborhood prompts that the model gets correct. In COUNTER­FACT, all such prompts have the same correct answer $o_i^c$:

$$\mathbb{E}_i \left[ \mathbb{E}_{p \in \text{neighborhood prompts}(s_i, r_i)} \left[ o_i^c = \operatorname*{argmax}_{x_E} \mathbb{P}_G \left[ x_E \mid p \right] \right] \right]. \tag{23}$$

We also report an aggregated **Score**: the harmonic mean of Efficacy, Paraphrase, and Specificity.

### C.2  FOR COUNTERFACT

COUNTERFACT contains an assortment of prompts and texts for evaluating model rewrites (Figure 14). This section provides formal definitions for each COUNTERFACT metric. First, the probability tests:

- **Efficacy Success** (**ES**) is the proportion of cases where $o_i$ exceeds $o_i^c$ in probability. Note that the prompt matches exactly what the edit method sees at runtime:

$$\mathbb{E}_i \left[ \mathbb{P}_G \left[ o_i \mid p(s_i, r_i) \right] > \mathbb{P}_G \left[ o_i^c \mid p(s_i, r_i) \right] \right]. \tag{24}$$

- **Paraphrase Success** (**PS**) is the proportion of cases where $o_i$ exceeds $o_i^c$ in probability on rephrasings of the original statement:

$$\mathbb{E}_i \left[ \mathbb{E}_{p \in \text{paraphrases}(s_i, r_i)} \left[ \mathbb{P}_G \left[ o_i \mid p \right] > \mathbb{P}_G \left[ o_i^c \mid p \right] \right] \right]. \tag{25}$$

- **Neighborhood Success** (**NS**) is the proportion of neighborhood prompts where the models assigns higher probability to the correct fact:

$$\mathbb{E}_i \left[ \mathbb{E}_{p \in \text{neighborhood prompts}(s_i, r_i)} \left[ \mathbb{P}_G \left[ o_i \mid p \right] < \mathbb{P}_G \left[ o_i^c \mid p \right] \right] \right]. \tag{26}$$

- **Editing Score** (**S**), is the harmonic mean of ES, PS, and NS.

Now, the generation tests:

- **Reference Score** (**RS**) measures the consistency of $G$'s free-form generations. To compute it, we first prompt $G$ with the subject $s$, then compute TF-IDF vectors for both $G(s)$ and a reference Wikipedia text about $o$; RS is defined as their cosine similarity. Intuitively, $G(s)$ will match better with $o$'s reference text if it has more consistent phrasing and vocabulary.

- We also check for excessive repetition (a common failure case with model editing) using **Generation Entropy** (**GE**), which relies on the entropy of $n$-gram distributions:

$$- \left( \frac{2}{3} \sum_k f_2(k) \log_2 f_2(k) + \frac{4}{3} \sum_k f_3(k) \log_2 f_3(k) \right). \tag{27}$$

Here, $f_n(\cdot)$ is the $n$-gram frequency distribution.

## D   EDITING DIFFERENT CATEGORIES OF FACTS TOGETHER

For an edit $(s, r, o)$, $r$ associates a subject $s$ and object $o$. Both $s$ and $o$ have their associated *types* $\tau(s)$ and $\tau(o)$. For example, $r =$ "is a citizen of" is an association between a `Person` and `Country`. We say that $\tau(s_1)$ and $s_2$ are *diverse* if $\tau(s_1) \neq (\tau(s_2))$, and *similar* otherwise. The definition follows similarly for objects. For any relation pair $(r_1, r_2)$, we sample from COUNTERFACT a set of edits $\mathcal{E}_{mix} = \{(s, r, o) \mid r \in \{r_1, r_2\}\}$, such that numbers of edits for each relation are equal. We compare MEMIT's performance on the set of edits $\mathcal{E}_{mix}$ in four pairs of relations that have different levels of diversity between them. Each relation is followed by its corresponding `relation_id` in WikiData:

(a) Subject different ($\tau(s_1) \neq \tau(s_2)$), Object different ($\tau(o_1) \neq \tau(o_2)$):

$$(\tau(s_1) = \text{Person}, r_1 = \text{citizen of } (\textbf{P27}), \tau(o_1) = \text{Country}),$$

$$(\tau(s_2) = \text{Country}, r_2 = \text{official language } (\textbf{P37}), \tau(o_2) = \text{Language})$$

(b) Subject similar ($\tau(s_1) = \tau(s_2)$), Object different ($\tau(o_1) \neq \tau(o_2)$):

$$(\tau(s_1) = \text{Person}, r_1 = \text{plays position in sport } (\textbf{P413}), \tau(o_1) = \text{Sport position}),$$

$$(\tau(s_2) = \text{Person}, r_2 = \text{native language } (\textbf{P1412}), \tau(o_2) = \text{Language})$$

(c) Subject different ($\tau(s_1) \neq \tau(s_2)$), Object similar ($o_1 = \tau(o_2)$):

$$(\tau(s_1) = \text{Place}, r_1 = \text{located in } (\textbf{P17}), \tau(o_1) = \text{Country}),$$

$$(\tau(s_2) = \text{Item/Product}, r_2 = \text{country of origin}(\textbf{P495}), \tau(o_2) = \text{Country})$$

(d) Subject similar ($\tau(s_1) = \tau(s_2)$), Object similar ($\tau(o_1) = \tau(o_2)$):

$$(\tau(s_1) = \text{Person}, r_1 = \text{citizen of } (\textbf{P27}), \tau(o_1) = \text{Country}),$$

$$(\tau(s_2) = \text{Person}, r_2 = \text{works in } (\textbf{P937}), \tau(o_2) = \text{City/Country})$$

Figure D depicts MEMIT rewrite performance in these four scenarios. We find that the effectiveness of $\mathcal{E}_{mix}$ closely follows the average of the individual splits. Therefore, the presence of diversity in the edits (or lack thereof) does not tangibly influence MEMIT's performance.

## E   DEMONSTRATIONS

This section provides two case studies, in which we apply MEMIT to mass-edit new or corrected memories into GPT-J (6B).

**Knowledge freshness.**   On November 8th, 2022, the United States held elections for 435 congressional seats, 36 governor seats, and 35 senator seats, several of which changed hands. We applied MEMIT to incorporate the election results into GPT-J in the form of (`congressperson, elected from, district`) and (`governor/senator, elected from, state`).[4] The MEMIT edit attained 100% efficacy (ES) and 94% generalization (PS).

**Application in a specialized knowldge domain.**   For a second application, we used MEMIT to create a model with specialized knowledge of amateur astronomy. We scraped the names of stars that were referenced more than 100 times from WikiData and belong to one of the 18 constellations named below.

| Andromeda, | Aquarius, | Cancer, | Cassiopeia, | Gemini, | Hercules, |
|---|---|---|---|---|---|
| Hydra, | Indus, | Leo, | Libra, | Orion, | Pegasus, |
| Perseus, | Pisces, | Sagittarius, | Ursa Major, | Ursa Minor, | Virgo |

We obtained 289 tuples of the form (`star, belongs to, constellation`). The accuracy of the unmodified GPT-J in recalling constellation of a star was only 53%. Post-MEMIT, accuracy increased to 86%.

---

[4]The results were available before November 14th.

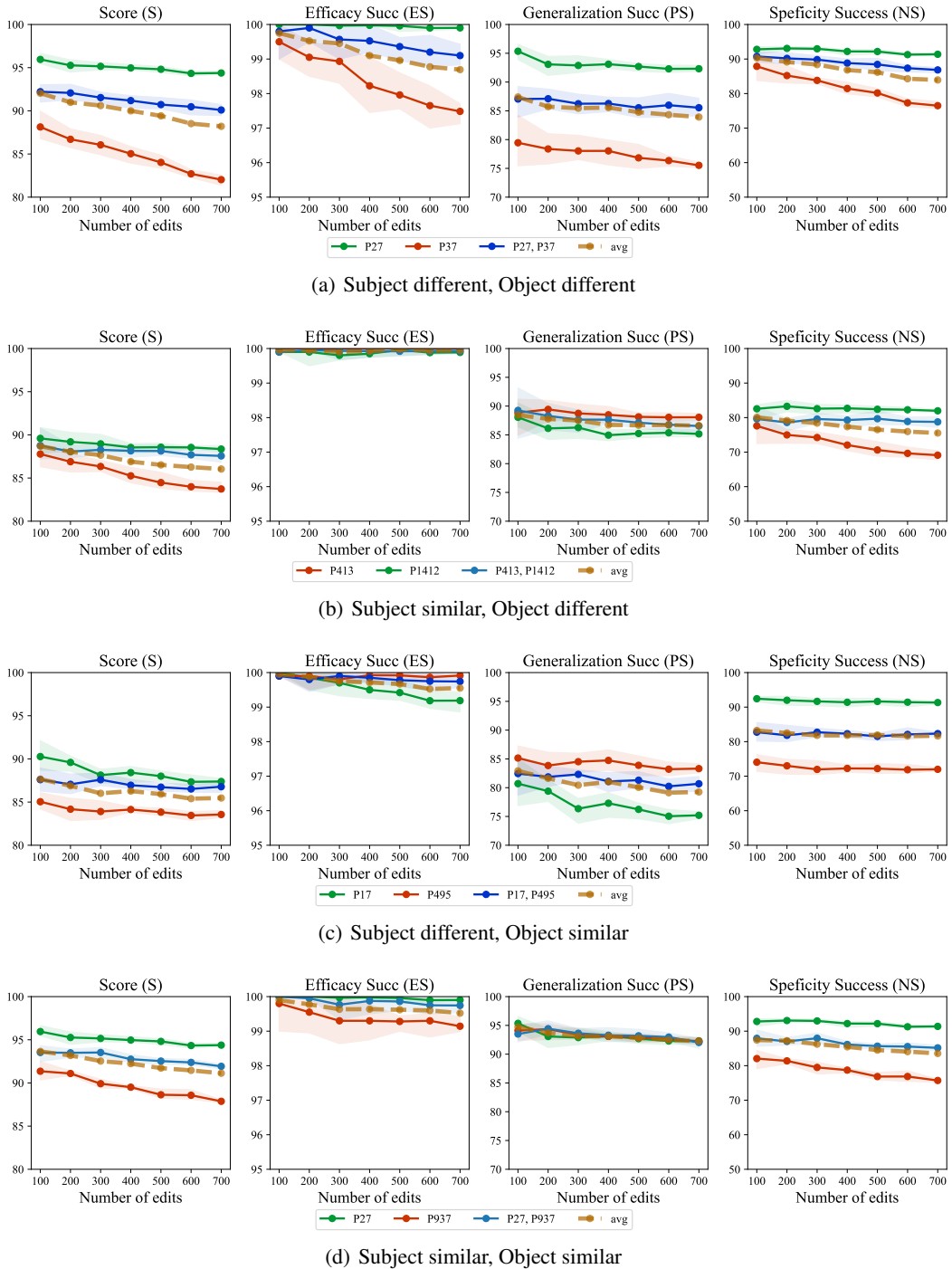

Figure 10: MEMIT's performance while editing memories with four levels of diversity. Each data point is a mean of 10 experiments. Filled areas show 90% confidence intervals of the values from those experiments.

# F   ABLATIONS

MEMIT contains several critical design choices: it uses a (i) range of critical mid-layer (ii) MLP modules at the (iii) last subject token, with the (iv) hyperparameter $\lambda$ (Eqn. 15) to control the impact of the update. Choice (iii) was already demonstrated by Meng et al. (2022) to be significant through an ablation study, but we now investigate the other three.

## F.1   VARYING THE NUMBER AND LOCATION OF EDITED LAYERS

We test five total configurations of $\mathcal{R}$, the set of critical MLP layers to be targeted during editing. Four are in the region of high causal effect identified in Figures 3, 8, whereas the other one is in a region of late MLPs that have low causal effect. As Figure 11 shows, using more layers yields higher efficacy and generalization while also improving specificity. Moreover, edits at the late-layer MLPs are considerably worse. These results confirm the importance of the causal analysis to MEMIT's performance.

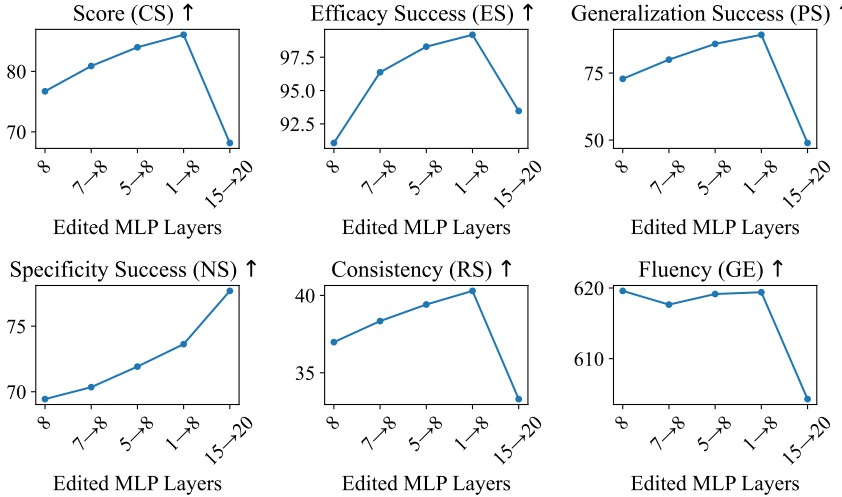

Figure 11: Varying the edited MLP layers

## F.2   VARYING THE TARGETED MODULE: EDITING ATTENTION

Next, we check whether edits at either early or late-layer attention modules perform comparably to their MLP counterparts. As Figure 12 shows, attention edits perform considerably worse.

## F.3   VARYING THE COVARIANCE HYPERPARAMETER $\lambda$

Finally, we investigate the impact of the covariance adjustment factor (denoted $\lambda$ in Eqn. 15) on performance; Figure 13 displays the results. Specificity and fluency increase monotonically with $\lambda$, indicating that higher $\lambda$ values preserve original model behavior. However, at the same time, efficacy and generalization fall when $\lambda$ is increased. We can see that around $\approx 10^4$, the aggregated score reaches a maximum.

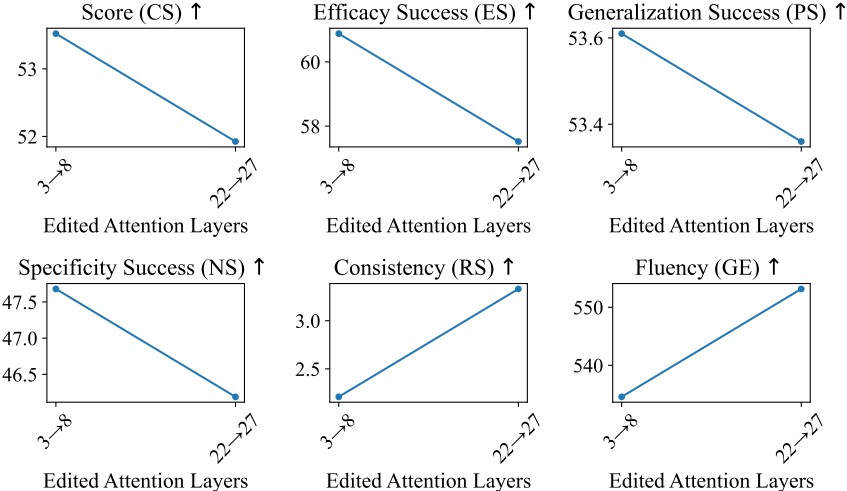

Figure 12: Varying the edited attention layers

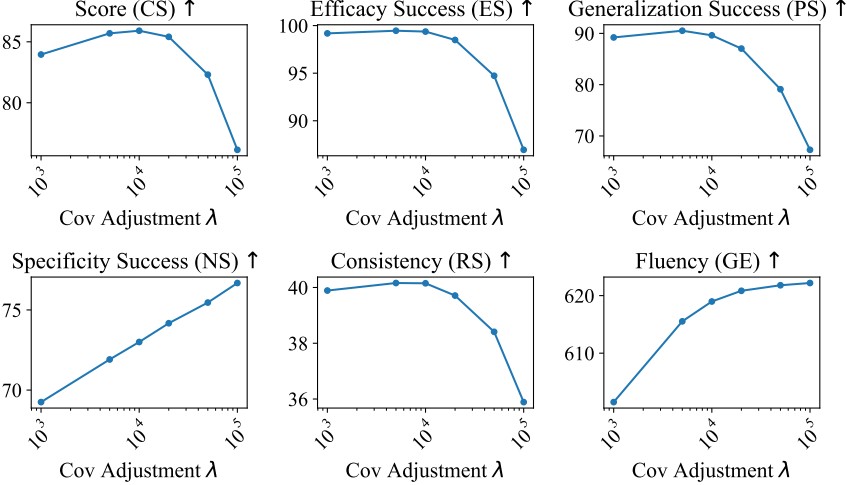

Figure 13: Varying the covariance adjustment factor $\lambda$

```
{
    "case_id":15,
    "pararel_idx":8062,
    "requested_rewrite":{
        "prompt":"{}, the",
        "relation_id":"P413",
        "target_new":{
            "str":"goaltender",
            "id":"Q1317534"
        },
        "target_true":{
            "str":"linebacker",
            "id":"Q528145"
        },
        "subject":"Percy Snow"
    },
    "paraphrase_prompts":[
        "Victoriano Salado Alvarez (1867-1931). Which position does Percy Snow play? They play as",
        "It is found in the Indomalayan realm. Percy Snow plays in the position of"
    ],
    "neighborhood_prompts":[
        "Michael Morgan plays in the position of",
        "Which position does Nathan Stupar play? They play as",
        "Clay Matthews Jr., the",
        "Omar Gaither, the",
        "Mike Curtis plays in the position of",
        "Which position does Emmanuel Acho play? They play as",
        "Clay Matthews Jr. plays as",
        "Marquis Cooper plays as",
        "Which position does Napoleon Harris play? They play as",
        "Brendon Ayanbadejo, the"
    ],
    "generation_prompts":[
        "Percy Snow is incredible at",
        "The expertise of Percy Snow becomes important when",
        "The expertise of Percy Snow becomes important when",
        "The expertise of Percy Snow becomes important when",
        "The expertise of Percy Snow becomes important when",
        "The expertise of Percy Snow becomes important when",
        "Percy Snow is incredible at",
        "The expertise of Percy Snow becomes important when",
        "The expertise of Percy Snow becomes important when",
        "Percy Snow is incredible at"
    ]
}
```

Figure 14: A sample of the COUNTERFACT dataset.

