# OpenReview forum: "Mass-Editing Memory in a Transformer"
_ICLR.cc/2023/Conference — ICLR 2023 notable top 25%_

### Official Review · Reviewer_6w9b · 2022-10-22

**Confidence:** 2
**Correctness:** 4
**Technical Novelty And Significance:** 2
**Empirical Novelty And Significance:** 3
**Recommendation:** 6

**Clarity, Quality, Novelty And Reproducibility:**

The proposed research problem is novel and seems useful.

The paper itself is opaque, stitching a sequence of steps together each of which is described without much insight. As an empirical work this is perhaps okay.

**Strength And Weaknesses:**

STRENGTHS
- A new problem proposal ("mass-editing") on the relatively new topic of memory editing in transformers
- On the proposed new setting, MEMIT is the only memory editing method that works.

WEAKNESSES
- I find the paper quite opaque. I find it difficult to learn much insight, besides the high-level impression that MEMIT extends ROME to make multiple edits and modify multiple layers at once. Why this is so necessary to achieve effective editing is not very clear, and consequently much of the work feels like engineering.

**Summary Of The Paper:**

The paper develops a batch update method for editing memory in transformers called MEMIT. Experiments are done on incorporating true/fake information (zsRE/CounterFact) using GPT-J and GPT-NeoX. On these tasks, MEMIT dramatically outperforms small update methods (MEND, ROME) when the number of edits is large.

**Summary Of The Review:**

The paper proposes MEMIT, a method for mass-editing memory in transformers, which works very well in the proposed setting.

---

> ### Author Response · Authors · 2022-11-16
> **Response to Reviewer 6w9b**
>
> Thank you for your review! We are glad that you found the MEMIT algorithm both novel and useful.
>
> ---
> > Why are multiple layers necessary to edit many facts at once?
>
> Recall that we view each MLP layer as an associative memory that stores $K = \left[ k_1, \dots, k_n \right] \rightarrow V = \left[ v_1, \dots, v_n \right]$ associations in its weight matrix $W \in \mathbb{R}^{R \times C}$, where $V \approx WK$.
>
> Crucially, matrices like $W$ have limited capacity. If we wanted lossless recall, we’d only be able to store $C$ facts in $W$, but even under the minimum squared error formulation, interference between facts will eventually take over as $n$ grows large.
>
> Fortunately, GPT models have many MLPs. As Section 4 describes, our hypothesis is that each MLP stores a *component* of the complete memory about any given subject. At each layer, a part of it is recalled, and the outputs are summed via the residual stream. Using multiple $W$s increases capacity; much of the legwork in Algorithm 1 is dedicated to taking advantage of more layers.
>
> We also see empirical evidence for the importance of multiple layers. We have added an ablation study to Appendix F, which demonstrates that reducing the number of edited layers or using the wrong layers significantly reduces performance.
>
> ---
>
> Hopefully this provides some intuition for the design choices in MEMIT! Please let us know if you’d like any other clarifications, or if you would consider updating your evaluation based on our response.

---

### Official Review · Reviewer_J5Nr · 2022-10-25

**Confidence:** 4
**Correctness:** 3
**Technical Novelty And Significance:** 3
**Empirical Novelty And Significance:** 3
**Recommendation:** 6

**Clarity, Quality, Novelty And Reproducibility:**

Clarity: The paper is quite well written.

Novelty: Moderate novelty. This paper iterates on previous work to perform editing on thousands of associations at once.

Quality: The technical proposal and the experiments are quite nice -- the knowledge editing seems quite limited in scope to be used for practical purposes.

Reproducibility: The promised code and data release should make it reproducible.

**Strength And Weaknesses:**

Pros:
- Knowledge editing seems quite intriguing especially for transformer models which seems like a black box. This seems to be a step in the right direction of model interpretability.


Cons:
- However, the knowledge that can be edited is based on associations related to subjects. I wonder if this can be extended to cover more general knowledge such as adding abilities for model to know a broad set of facts in a topic, etc.
- This methodology seems to rely on critical layers — do they always exist? what if all layers are critical? would the method scale to such behavior?
- The method also seems heavily based on 'last subject token' which seems quite specific.




**Summary Of The Paper:**

This paper proposes a methodology to edit memory in a transformer model which can be used to modify or add the model's knowledge.

**Summary Of The Review:**

The paper probes an interesting aspect of transformers language model, that is, how to modify knowledge encapsulated in the weights. This paper demonstrates that it is possible to do so in the factual association setting, which shows premise and shed some light on the interpretability. The scale of the edits is impressive compared to previous work. The experiments are quite well executed. However, the scope of this work for real-world applicability is still in question, as the knowledge editing pertains to explicit relations, or what the user calls associations.

---

> ### Author Response · Authors · 2022-11-16
> **Response to Reviewer J5Nr**
>
> Thank you for your review! We’re glad that you found our paper intriguing, novel, and well written.
>
> ---
> > Do “critical layers” always exist? Are facts always recalled at the “last subject token”?
>
> As Meng et al. 2022 observed, there are exceptions to both rules in pre-trained models:
> * When the model is not confident about the correct answer (i.e., $\mathbb{P}\left[ o \right] \ll 1$), we don’t see critical layers. In these cases, the model may not have the fact stored at all.
> * GPT will sometimes recall a memory *earlier* than the last subject token, if it can guess the answer based on previous words. For example, “Madame Bovary” is French, but the model can guess that without looking at “Bovary,” simply from the first token “Madame.”
>
> When editing a fact for a specific subject, we simply fix the set of critical layers $\mathcal{R}$ based on aggregated measurements of facts known by the model, and insert facts using the actual last subject token. This approach will replace the old fact if it already existed, and simply insert the new one if not.
>
> ---
> >  What if all layers are critical, does the method scale?
>
> In principle, Algorithm 1 could be extended to an arbitrary number of critical layers. However, we haven’t tried this because Causal Tracing reveals that there are only a few critical layers early in GPT-J/NeoX.
>
> ---
> > Does editing factual associations have real-world applicability?
>
> Yes! Language models are increasingly being used as implicit knowledge bases, but they can lack specialized knowledge and contain a lot of incorrect information. Retraining large models is infeasible (GPT-3 training is estimated to have consumed 1200MWh), so we would like to update facts in a more efficient way. See our first response to Reviewer Nkci for some examples.
>
> To demonstrate the speed and simplicity of applying MEMIT to a realistic use case, we used MEMIT to load the new members of U.S. congress from the 2022 midterm elections into GPT-J (6B), which was trained on The Pile, a dataset from 2020. We also loaded star names and constellations into a GPT-J model, increasing the accuracy of the model at recalling those facts. These demonstrations are summarized in Appendix E.
>
> ---
> > Can MEMIT be extended to more general knowledge?
>
> The key-value associations edited by MEMIT might encode other types of knowledge such as linguistic rules or common sense. That’s a promising direction for future work, but the extension to more abstract forms of knowledge may not be trivial. Our current work focuses only on factual $(s, r, o)$ associations.
>
> ---
>
> Thanks for all the great questions and suggestions! Please let us know if you have any remaining concerns, or if you would consider updating your evaluation based on our response.

---

### Official Review · Reviewer_NCki · 2022-10-25

**Confidence:** 4
**Correctness:** 3
**Technical Novelty And Significance:** 2
**Empirical Novelty And Significance:** 3
**Recommendation:** 8

**Clarity, Quality, Novelty And Reproducibility:**


**Clarity and quality:** overall, the paper is well-written and easy to follow, with a fairly standard overall structure. Subjectively, I would argue that it lacks practical motivation / justification, explaining when / why does one need to perform many edits in parallel.

**Novelty:** to the best of my understanding, the proposed algorithm is (1) novel and (2) first to tackle the specific problem (mass edits) better than FT-W. However, unless authors convince me otherwise, I believe that this is a fairly niche problem, limiting the potential impact.

**Reproducibility:** as i stated earlier, authors made considerable effort in making this paper reproducible, providing both configuration, instrutions, and the specific versions of all the libraries. If there was a score for reproducibility, I would rank it as high, definitely above average among the last year's accepted papers.

**Strength And Weaknesses:**



### Strengths

- to the best of my knowledge, this work is technically sound: the MEMIT algorithm appears valid and agrees with the current understanding of large language models
- the main evaluations deal with multi-billion-parameter language models - the types of models that are (arguably) most likely to need editing, since retraining them from scratch is extremely expensive
- authors take a considerable effort to make their experiments reproducible. Two rounds of applause for (1) writing a decent README and (2) specifying versions of all dependencies (in a conda yml).
- authors clearly state the main limitations of their work (with one major exception, see weaknesses)
- the paper is generally well-written and easy to follow

### Weakness

- Motivation: the paper focuses on *how* to to make $10^4$ edits, but talks very little about *why* would one need to do that. I can come up with some obvious use cases, e.g. running public machine translation systems that deal with bulk user feedback -- or patching GPT-J (2021) with all major news from 2022. However, I'd argue that it would be best if authors discuss these potential applications somewhere within the first 2 pages.

- Hiding runtime numbers: to the best of my udnerstanding, running MEMIT requires at least 0.9 hours per experiment (Appendix B.4), compared to <=2 minutes for MEND. While it does not invalidate the main contributions (and it is still faster than, say, ROME), it would be best to clearly state that the method is slower in the limitations section and in the main paper.

- Lacking ablation: MEMIT is a fairly complex algorithm with many design decisions. It would be best to validate that these decisions actually matter. How significant is it that MEMIT edits MLP (and not attention) layers? Is MEMIT sensitive to \lambda - and how sensitive? How does the efficacy / consistency change with the number of layers (i.e. if we restrict MEMIT to edit a smaller number of layers)?



### Missing related work

- https://openreview.net/forum?id=HJedXaEtvS - to the best of my knowledge, this is the first published work on editing transformers, though it is closer to MEND and likely not scalable to LLMs

### Questions

- in Table 1, does MEND perform all 10^4 edits sequentially, in parallel, or some combination of the two (e.g. $10$ times $10^3$)? (S5.1 correctly states that MEND *can* apply edits in parallel, but does not state if it does

- How many parameters, on average, are updated in MEMIT? (depending on the number of samples and the model) Is it a negligible fraction, or a significant potion of the entire model? Why asking: is it feasible to maintain multiple MEMIT updates in parallel like adapters / soft prompts?

- In appendix B.4, you state "If all such vectors are already computed, MEMIT takes 3226.35 sec". How long does it take to compute these vectors?


> GPT-NeoX runs require at least two: one 48GB GPU for running the model in float16, and another slightly smaller GPU for executing the editing method.

Are there any obvious ways to run MEMIT in a smaller memory footprint? E.g. does MEMIT workload allow for efficient RAM offloading?

**Summary Of The Paper:**

The paper studies the problem of editing factual knowledge in transformer-based language models. Authors note that existing model editing techniques fail when introducing a large batch of edits. They present a fairly model-specific algorithm called MEMIT, directly modifies model weights of FFN (MLP) transformer layers that matter most for a specific input. On GPT-J-6B and GPT-NEO-20B, MEMIT can perform a large batch (e.g. $10^4$) edits with reasonable efficacy and specificity.

**Summary Of The Review:**

To the best of my understanding, this paper offers a technically sound and practically effective solution to a relatively niche problem. While it has a few areas for improvement (e.g. explicitly compare algorithm runtime in the main paper, add clearer motivation in the first 2 pages). However, these changes are relatively minor, i.e. possible to implement within the author response period.

My overall recommendation is "Weak Accept" since this paper brings meaningful contribution to a niche audience. I can be convinced to raise my score if authors convince me that their work has (significantly) higher impact on researchers / practitioners than i currently assume.

---

> ### Author Response · Authors · 2022-11-16
> **Response to Reviewer NCki (Part 1/2)**
>
> Thank you for your review! We’re glad that you found MEMIT novel + meaningful, and we appreciate the shout-out about reproducibility.
>
> ---
> > *Why* would we want to make $10^4$ edits? What potential applications are there?
>
> Language models are often applied in contexts where we might want to efficiently update obsolete facts, correct mistakes, or add specialized knowledge. A few examples:
> * **Question answering & knowledge search**: LMs can be viewed as implicit knowledge bases, but they often contain incorrect facts and can quickly grow outdated. If LMs are ever used as search tools, we will want to fix these problems and perhaps add specialized knowledge for specific use-cases (e.g., treatments to diseases).
> * **Creative writing**: LMs are often used to aid humans in writing stories or dreaming up fantastical alternate worlds. Knowledge editing can be used to teach the model about new characters or change other facts about the new world.
> * **Generating marketing content**: As businesses increasingly adopt LMs for creating written content, it is important for the models to know about the business’s products or services. Companies can use knowledge editing to customize LMs to their specific product and solution space.
> * **Receiving & incorporating user feedback**: In any context where user feedback is collected (e.g., machine translation or grammar checkers), we want methods for directly incorporating feedback into the model.
>
> These scenarios span a wide application space, so we think scalable model editing is an important problem. We’ve added a discussion about motivation to the introduction.
>
> To further support our argument, we have added two practical demonstrations of MEMIT to Appendix E.
> * During the rebuttal period, the United States held elections for 435 congressional seats, 36 governor seats and 35 senator seats, several of which changed hands. We used MEMIT to incorporate the new election results into GPT-J in the form of `(congressperson, elected from, district)` and `(governor/senator, elected from, state)`. The measured efficacy score (ES) of the rewrite was 100%, and the generalization score (PS) was 94%.
> * For a second application, we used MEMIT to create a model with specialized knowledge of amateur astronomy. We demonstrate that MEMIT can be used to improve GPT-J’s accuracy in recalling the constellation associated with a star.
>
> ---
> > Ablations: which design choices actually matter?
>
> Thank you for the suggestion! We have added an ablation study to Appendix F, which documents the effect of varying several design choices. In summary:
> * Editing the late-layer attention (red regions of high causal effect in Figure 8c) results in poor performance.
> * Reducing the number of layers hurts all of efficacy, generalization, and specificity. Using the incorrect MLPs (late-layer ones with low causal effect) performs poorly.
> * When $\lambda$ (the scaling factor in Eqn. 15) is too low, model damage is high (specificity and fluency drop). Conversely, when it is too high, efficacy and generalization drop. Our choice of $1.5 \times 10^4$ achieves a good balance.
>
> ---
> > It would be best to clearly state that the method is slower in the limitations section and in the main paper.
>
> That is an important point. We have:
> * Clarified MEMIT’s slower runtime as a limitation in the results section.
> * Reported runtime numbers in the main paper.
> * Clearly highlighted that runtime details are in Appendix B.
>
> ---
> > Does MEND perform all $10^4$ edits sequentially, in parallel, or some combination of the two?
>
> MEND’s edits are fully parallel, in accordance with the official implementation provided by Mitchell et al. 2021. The gradients from all inputs are accumulated, then passed through the hypernetwork together. We have adjusted Section 5.1 and Appendix B.2 to clarify that.
>
> ---
> > How many parameters are updated in MEMIT? Is it feasible to maintain multiple MEMIT updates in parallel like adapters / soft prompts?
>
> MEMIT updates all the fan-in MLP layers in the critical range $\mathcal{R}$. That amounts to $|\mathcal{R}| * \mathrm{in\_dim} * \mathrm{out\_dim}$ weights:
> * In GPT-J, MEMIT edits $1.6$ GB of `fp32` weights, which is about 6.5% of the total model size.
> * In GPT-NeoX, MEMIT edits $1.8$ GB of `fp16` weights, which is about 4.5% of the total model size.
>
> That means we can efficiently store multiple updates in parallel (as you can do with adapters or soft prompts), while consuming much less memory than cloning the entire model. In practice, in deployment, do use this technique. Dynamically swapping between sets of MEMIT-updated matrices allows us to store different models customized with different knowledge with only incremental RAM use.
>
> ---
> Response continued below (1/2)

---

> > ### Author Response · Authors · 2022-11-16
> > **Response to Reviewer NCki (Part 2/2)**
> >
> > ---
> > > How long does it take to compute the $z_i$ vectors that are inserted with MEMIT?
> >
> > It takes $6.54$ hours to compute $10{,}000$ vectors. However, the current implementation is naive and does not batch $z_i$ optimizations, which are independent and actually “embarrassingly parallel.” We have not yet benchmarked a batched implementation, but we will include one in the final version if accepted. We have added some comments about this to Appendix B.
> >
> > ---
> > > Are there any obvious ways to run MEMIT in a smaller memory footprint?
> >
> > MEMIT’s memory bottleneck is the inversion of a large matrix (Algorithm 1, line 14). $(C + KK^T)^{-1}$ is $\approx 16{,}000^2$ in GPT-J (6B), and $\approx 24{,}000^2$ in GPT-NeoX (20B). We might be able to bring memory usage down by applying approximations, but it does not seem very necessary, as a $24{,}000^2$ matrix in `fp32` is only around 2.3 GB.
> >
> > The reason we need two GPUs for GPT-NeoX is not necessarily due to MEMIT’s memory usage itself, but rather that NeoX consumes all memory available on our GPU hardware even in `fp16`.
> >
> > ---
> > > Missing related work
> >
> > Thank you for noticing this issue; we have added the citation to Sinitsin, et al.
> >
> > ---
> >
> > Thanks for all the great questions and suggestions! Please let us know if you have any remaining concerns, or if you would consider updating your evaluation based on our response.

---

> > > ### Comment · Reviewer_NCki · 2022-11-22
> > > **Response**
> > >
> > > I thank authors for answering my questions. I am incrementing my score based on the revised version of the paper and the requested parts (motivation, ablation, discussing memory, etc).

---

### Official Review · Reviewer_55ot · 2022-10-25

**Confidence:** 4
**Correctness:** 4
**Technical Novelty And Significance:** 3
**Empirical Novelty And Significance:** 3
**Recommendation:** 8

**Clarity, Quality, Novelty And Reproducibility:**

Clarity: the work was clearly written and presented.

Quality: the paper is well-written, the model choices well-motivated and explained, and the experiments appear to be sound.

Novelty: while drawing on some previous work on causal intervention, this is a clearly novel proposal that has notable scaling results that have not been seen in previous work in this area. This is an important contribution in a newly developing area of research.

Reproducibility: while I did not thoroughly review the code provided in the supplementary material, the authors appear to have shared the details needed to reproduce their method and results. The authors also plan to open-source the method after publication.


**Strength And Weaknesses:**

Strengths
- Well-motivated by previous literature and views on memory in transformers/NNs
- Thorough and clear section to explain the editing algorithm (although see question below)
- Experimental results show a clear advance over previous methods for large-scale fact editing in transformer architectures
    - Experiments include a comparison to a fine-tuning baseline
- Clearly notes limitations of their method (i.e. doesn't cover spatial or temporal reasoning)
- Authors plan to open-source the method on publication

Weaknesses
- Needs more clarity on the evaluation metrics used in the experiments
    - The metrics in Table 1 are presented in 5.2.1, and several more metrics in 5.2.2. Are these actually the same metrics with slightly different names? Some specificity on this would be appreciated.

Further questions
- In Eq. 17 and in Algorithm 1, line 6, the layers are $l \in \mathcal{S}$. However, in section 4.1, Figure 2, and the beginning of 4.3, the critical layers are denoted $l \in \mathcal{R}$. Is this difference a mistake, or is there a difference between $\mathcal{S}$ and $\mathcal{R}$? If there is, please define $\mathcal{S}$ in 4.3, or somewhere near Eq. 17, to make this clear.

**Summary Of The Paper:**

This work addresses the problem of updating world knowledge in large language models (LLMs), which learn many common facts through training on large-scale datasets. This problem is an important one to the machine learning community, as many applications rely on knowledge acquired by LLMs for different applications. The authors narrow down the problem to (subject, relation, object) triplets, or ($s_i, r_i, o_i$). This is assessed by giving the model a natural language prompt $p_i = p(s_i, r_i)$ and having the LLM predict $o_i$. The edits keep subject $s_i$ and relation $r_i$ the same, but modify the associated object $o_i$. This is accomplished by analyzing and modifying the feedforward MLP layers of a transformer, which have been shown to contain key-value memories.

First, the authors identify which layers are critical to storing a given fact, using a causal tracing method. They then use the insight that the transformer MLP stores key-value memories to estimate a change $\delta$ that updates a given layer's MLP to store the updated memory (in the form of an ($s_i, r_i, o_i$) triplet). Each layer in the set of critical layers is altered to contribute an approximately equal portion of the overall change needed for the model to output the updated ($s_i, r_i, o_i$). The estimated updates are found by using gradient descent to move the hidden state representations toward responding with the updated object $o_i$ in response to the prompt $p_i$. The overall method is called MEMIT.

The authors evaluate their method by measuring several metrics to test how well the model has learned the new edited facts, measuring efficacy, generalization, specificity, fluency, and consistency of its responses. This is done by evaluating how well MEMIT preforms on adding correct information, as well as adding counterfactual information, which is a false fact that has a low score from the original unedited LLM. These are datasets used in a recent work (Meng et al. 2022). Their results show that MEMIT performs especially well at scaling the number of edits compared to other recent editing methods.

**Summary Of The Review:**

Overall, I enjoyed reading this paper and found the contributions clear and the results convincing. While I am not intimately familiar with the recent literature on model editing, this is clearly an important problem that seems to be well-addressed by this method. In particular, the success of their approach in scaling to thousands of edits makes it a promising proposal for modifying LLMs over time. I also appreciate that the authors discuss the limitations of this approach as well -- it is restricted to working with $(s, r, o)$ triplets, and does not cover many other types of knowledge that LLMs seem to acquire. However, the method may provide a template for ways to address other types of declarative knowledge.

---

> ### Author Response · Authors · 2022-11-16
> **Response to Reviewer 55ot**
>
> Thank you for your review! We’re glad that you found our methods well motivated, important, and convincing.
>
> ---
> > Clarity on evaluation metrics, particularly the difference between those in 5.2.1 and 5.2.2.
>
> Yes: the metrics in 5.2.1 (for zsRE) and 5.2.2 (for CounterFact) are different. The zsRE metrics measure the proportion of cases where the target answer $o_i$ is *top-1 correct*, whereas CounterFact metrics test for the target answer $o_i$ having *greater probability* than the ground truth answer $o_i^c$ (recall that CounterFact consists of *false* facts, each corresponding to a ground truth one that we want to overwrite).
>
> These metric choices are consistent with Meng et al. 2022. They adopted the zsRE metrics from past works but designed CounterFact separately, citing two major reasons for switching to probability comparisons:
> * Accuracy is **less sensitive**, because it discards probability information. For example, the specificity accuracies of FT+W, MEND, and MEMIT are all approximately equal in Table 1, despite the methods having very different quantitative and qualitative behavior in reality.
> * Accuracy is **too strict**. For an edit to be effective, $o_i$ doesn’t necessarily have to be the argmax. To maintain grammatical fluency, models may first utter helper verbs and other syntactic structures before generating the factual prediction.
>
> We have further clarified the difference between these two cases Appendix C.
>
> ---
> > Is there a difference between S and R?
>
> Thanks for catching this typo! $\mathcal{R}$ is the correct variable for the set of targeted layers. We have adjusted Algorithm 1 and Eqn. 17 accordingly.

---

### Decision · Program_Chairs · 2023-01-20

**Decision:**

Accept: notable-top-25%

**Justification For Why Not Higher Score:**

As pointed by the Reviewer 6w9b, much of the work employs engineering techniques.


**Justification For Why Not Lower Score:**

All the reviewers gave positive reviews.

**Metareview: Summary, Strengths And Weaknesses:**

This paper studies the problem of updating factual knowledge in large language models (LLMs). This is a new and well-motivated problem as all of us agree. The proposed method (MEMIT) for this problem is technically sound and empirically works well. Therefore all the reviewers vote for the acceptance.

The reviewers raised some concerns, including the unclear choices in the method design, clarity on the evaluation metrics and missing analysis like ablation study. During the rebuttal period,  most of the raised concerns have been properly addresses. There are no major concerns left.

**Note From Pc:**

if the above contains the word "oral" or "spotlight" please see: "oral" presentation means -> notable-top-5% and "spotlight" means -> notable-top-25%. As stated in our emails, we are disassociating presentation type from AC recommendations